# Examining heterogeneity in dementia using data-driven unsupervised clustering of cognitive profiles

Sayantan Kumar[1,2]*, Inez Y. Oh[2], Suzanne E. Schindler[3], Nupur Ghoshal[3,4], Zachary Abrams[2], Philip R. O. Payne[1,2]

**1** Department of Computer Science and Engineering, McKelvey School of Engineering, Washington University in St Louis, St. Louis, Missouri, United States of America, **2** Institute for Informatics, Data Science and Biostatistics (I2DB), Washington University School of Medicine, St. Louis, Missouri, United States of America, **3** Division of Neurology, Washington University School of Medicine, St Louis, Missouri, United States of America, **4** Department of Psychiatry, Washington University School of Medicine, St Louis, Missouri, United States of America

* sayantan.kumar@wustl.edu

**Data Availability Statement:** All relevant data are within the paper and its Supporting Information files.

## Abstract

Dementia is characterized by a decline in memory and thinking that is significant enough to impair function in activities of daily living. Patients seen in dementia specialty clinics are highly heterogenous with a variety of different symptoms that progress at different rates. Recent research has focused on finding data-driven subtypes for revealing new insights into dementia's underlying heterogeneity, rather than assuming that the cohort is homogenous. However, current studies on dementia subtyping have the following limitations: (i) focusing on AD-related dementia only and not examining heterogeneity within dementia as a whole, (ii) using only cross-sectional baseline visit information for clustering and (iii) predominantly relying on expensive imaging biomarkers as features for clustering. In this study, we seek to overcome such limitations, using a data-driven unsupervised clustering algorithm named SillyPutty, in combination with hierarchical clustering on cognitive assessment scores to estimate subtypes within a real-world clinical dementia cohort. We use a longitudinal patient data set for our clustering analysis, instead of relying only on baseline visits, allowing us to explore the ongoing temporal relationship between subtypes and disease progression over time. Results showed that subtypes with very mild or mild dementia were more heterogenous in their cognitive profiles and risk of disease progression.

## 1. Introduction

Dementia is characterized by a decline in memory and thinking, that is significant enough to impair function in activities of daily living. It can be caused by reversible causes such as medication-induced cognitive impairment, and irreversible conditions such as progressive neuro-degenerative conditions [1–3]. Alzheimer's Disease (AD) is the most common cause of dementia in older adults, and other common causes include cerebrovascular disease and

**Funding:** The preparation of this manuscript was supported by the Centene Corporation contract (P19-00559) for the Washington University-Centene ARCH Personalized Medicine Initiative. There was no additional external funding received for this study.

**Competing interests:** The authors have declared that no competing interests exist.

disorders linked to Lewy bodies, tau tangles, or limbic-predominant age-related TDP-43 encephalopathy (LATE) [4]. Notably, many patients have multiple conditions contributing to their cognitive impairment [5]. Dementia patients exhibit diverse symptoms and progress at varying rates, likely influenced by underlying brain pathologies, baseline cognitive ability, genetic factors, medical conditions, and social determinants of health. Consequently, patients treated in dementia clinics manifest significant heterogeneity, representing a spectrum of dementia subtypes [6–8]. A more thorough understanding of this clinical heterogeneity could enhance dementia diagnosis and prognosis, facilitating tailored care for patients and their families [9, 10].

Many neurological diseases are characterized by neuropathological features. Previous studies concerning the derivation of dementia types have frequently relied on suspected neuro-pathological diagnoses based on clinical features rather than objective data-driven methods [11–15]. Recently, the widespread availability of Electronic Health Records (EHR) data alongside evolving machine learning techniques has enabled data-driven approaches to reveal new insights into dementia's underlying heterogeneity [16]. For instance, clustering algorithms can stratify dementia patients into subtypes based on key EHR features, enhancing predictive ability compared to analyzing the entire cohort as a single homogeneous group [17–19]. However, current research on dementia subtyping faces three main limitations. First, there is a significant focus on parsing heterogeneity within AD-related dementia potentially neglecting important insights that could arise from examining heterogeneity within all-cause dementia as a whole [20]. Second, conventional clustering methods used in dementia research have analyzed cross-sectional single time-point data; however, given the heterogeneity in disease progression, mapping longitudinal trajectories is an important focus for dementia research [21–23]. Finally, existing research predominantly relies on imaging biomarkers obtained through expensive procedures, such as magnetic resonance imaging (MRI) and positron emission tomography (PET) scans. However, current research often overlooks the potential of using non-imaging clinical data, such as routinely collected electronic health records (EHR), as a valuable, cost-effective, and non-invasive solution for addressing heterogeneity in dementia [16, 24, 25]. Addressing these gaps is essential for effective clinical decision-making and precision diagnostics tailored to each subtype.

In response to the gaps in knowledge described above, our goal via this project was to analyze the heterogeneity of cognitive performance and disease progression within a real-world clinical dementia cohort. Towards this end, we aimed to delineate subtypes using a newly-developed unsupervised clustering technique called SillyPutty on cognitive assessment scores of patients seen in a memory clinic (Fig 1). This new heuristic clustering approach starts with a set of clustering assignments obtained from hierarchical clustering and iteratively adjusts cluster assignments to maximize the average silhouette width [26]. A novel aspect of our approach was the inclusion of all longitudinal patient visits for clustering analysis, rather than solely relying on baseline visits, which allowed us to examine the longitudinal relationship between subtypes and disease progression. Specifically, we aimed to (i) examine how subtypes at similar stages of dementia differ in their cognitive characteristics (ii) analyze patient transitions between different subtypes across multiple visits to examine the heterogeneity (variability) in progression rate amongst patients at different stages of dementia (Fig 1).

## 2. Materials and methods

### 2.1 Data sources and study participants

This retrospective study analyzed electronic health records (EHR) data extracted from the Washington University in St. Louis Research Data Core (RDC), a repository of patient clinical

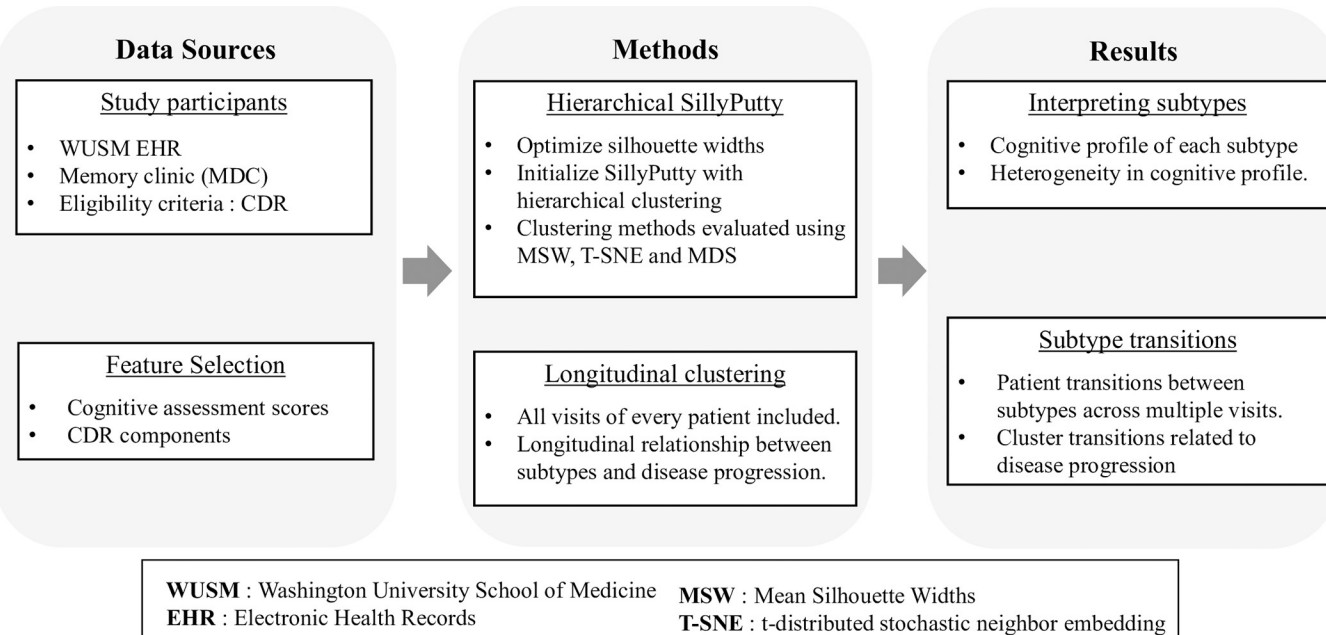

**Fig 1. Study workflow.** Outpatient visits at the Memory Diagnostic Center with a Clinical Dementia Rating recorded in the Washington University School of Medicine Electronic Health Records were included. Feature selection was performed to identify the optimal set of features for clustering. The SillyPutty algorithm [27] with hierarchical clustering was used to identify clusters from longitudinal patient visits. Identified clusters were analyzed for variability (heterogeneity) in cognitive characteristics. Patient transitions between subtypes across multiple visits were examined for variability in rate of disease progression.

data sourced from BJC HealthCare and Washington University Physicians. Approval for this study was obtained from the Washington University Institutional Review Board (IRB # 201905161), which granted a waiver of HIPAA Authorization for the use of Protected Health Information (PHI). Data for this study was accessed on November 20, 2020. The study cohort included all patients treated at the Memory Diagnostic Center (MDC) at Washington University School of Medicine (WUSM) for evaluation of memory and/or thinking concerns (Fig 1). The dataset, sourced from Allscripts TouchWorks, included office visits with cognitive assessment measures recorded between June 1, 2013, and May 31, 2018. This timeframe was chosen to mitigate potential data harmonization challenges arising from an EHR system transition that commenced on June 1, 2018.

All patients treated at the WUSM MDC underwent a comprehensive history and neurologic examination. A trained medical assistant also administered the cognitive assessment battery, including the Boston Naming Test [28], Mini-Mental State Exam [29], Short Blessed [30], Word List Memory Task [31], and Verbal Fluency [32], to most patients. The cognitive battery was often not performed in patients with moderate to severe dementia, who may not be able to complete the cognitive battery or who are distressed by testing. Memory specialists utilize the Clinical Dementia Rating (CDR®) scale to evaluate dementia severity, where scores ranged from 0 (normal cognition) to 3 (severe dementia). The CDR assess intra-individual changes in memory, thinking, and function compared to previous abilities [33]. Notably, memory specialists used the results of the cognitive assessment battery in the formulation of the CDR and documented them in the patient's EHR. Longitudinal data from 1,845 patients with 2,737 visits were eligible for inclusion, where each visit recorded a global CDR score (Fig 1).

## 2.2 Feature selection and preprocessing

**2.2.1 Feature selection for clustering.** We performed a qualitative feature selection step to determine the optimal set of features for clustering. We started with two sets of features as follows: (i) Cognitive assessment scores: Boston Naming Test [28], Mini-Mental State Exam [29], Short Blessed [30], Word List Memory Task [31], Verbal Fluency [32] and (ii) six components of the CDR score: Memory, Orientation, Judgment and Problem Solving, Community Affairs, Home and Hobbies, and Personal Care. For details regarding the range and interpretation of each of these tests, see S1 Table. The CDR components assess performance in six cognitive and functional domains, providing critical insights into different aspects of a patient's cognitive health respectively. The component scores include the values 0, 0.5, 1, 2 and 3 with 0 indicating normal cognition and 3 indicating severe impairment. Following pre-defined scoring rules as described in Morris et al., [33] the component scores can be aggregated to form the CDR score. We refer to the aggregated CDR as the global CDR score throughout the remainder of the manuscript, to avoid confusion with the CDR components. The *t*-distributed stochastic neighbour embedding (T-SNE) distribution and the Principal Component Analysis (PCA) for the 2 feature sets were analyzed to determine the optimal set of features for clustering (Fig 1) [34].

**2.2.2 Feature pre-processing.** Feature pre-processing included clipping the outlier values to the 5th and 95th percentile values and scaling between [0,1] using the Minmax Scalar package from sklearn (version 1.5.0) [35]. Since each of the selected features are ordinal variables and had a very low missing rate of <5%, we imputed the missing values for each feature column using the median value of that feature across all visits of all patients, following previous work [36] (Fig 1).

## 2.3 Unsupervised clustering

**2.3.1 SillyPutty algorithm.** SillyPutty is a heuristic clustering method that optimizes cluster assignments using silhouette widths. The silhouette width, which ranges from -1 to 1, is a metric that indicates the quality of a data point's assignment to its cluster. A value close to 1 signifies strong clustering, with distinct separation between clusters and high cohesion within each cluster. A value near 0 implies overlapping clusters, while a value close to -1 suggests the data point may be misclassified.

The goal of SillyPutty is to maximize the average silhouette width by iteratively refining cluster assignments. It begins with an initial set of clusters, which can be user-defined, randomly chosen, or derived from other methods. In each iteration, the algorithm calculates silhouette widths for the current clusters. It then identifies the data point with the lowest silhouette width and reassigns it to the nearest cluster. This process continues until all points have non-negative silhouette widths or until early termination conditions are met, such as reaching the maximum number of iterations (N = 100) or detecting a repeated silhouette width vector within a user-specified number of iterations. The final output includes the refined clusters, silhouette widths, and additional relevant information.

**2.3.2 SillyPutty with hierarchical clustering.** The standalone SillyPutty algorithm starts with purely random cluster assignments, repeating the algorithm with different random starting points (Random SillyPutty). Since SillyPutty can start with any cluster assignments, we initialized the cluster assignments with hierarchical clustering before applying the SillyPutty algorithm (hierarchical SillyPutty). This choice is motivated by results in the original SillyPutty paper where hierarchical SillyPutty outperformed other clustering methods such as Random SillyPutty, Partition Against Medoids (PAM) and hierarchical clustering on multiple simulated datasets.

**2.3.3 Clustering evaluation and baselines.** The optimal number of clusters for the Hierarchical SillyPutty algorithm was determined by the best mean silhouette width, for a range of clusters values (K) from K = 2 to 16. Hierarchical SillyPutty was compared with alternative clustering techniques: (i) Random SillyPutty (SillyPutty with random initial assignments), (ii) hierarchical clustering, (iii) Partition Against Medoids (PAM), (iv) K-Means and (v) DBSCAN. For a fair comparison, the same number of clusters selected for hierarchical SillyPutty was also used for the baseline methods. Further, all clustering methods were qualitatively evaluated using the metrics used in the original SillyPutty paper, such as Mean Silhouette Widths (MSW) [27, 37], T-SNE [34] and multidimensional scaling (MDS) [38].

## 2.4 Clustering on longitudinal visits

In our study, clustering analyses were performed on all data available from a 5-year period between 2013–2018. Note that our cohort included both patients with a single visit and patients with multiple visits during this period. For the clustering step, each visit was assumed independent without any temporality (linkage) between individual visits of the same patient. The temporality information between individual visits was used subsequently for analyzing the relationship between different clusters. This approach–including all visits for clustering instead of using only the baseline visit of each patient was taken to enable further longitudinal analysis and track the symptom progression rate of patients. For example, at any given point in time, a patient exists in a single cluster (dementia subtype) but may transition between different clusters over time (Fig 1). To gain an insight into which patients have a higher probability of progression to more severe dementia, patient transitions between different dementia subtypes were analyzed across multiple visits. Finally, the differences in progression rate, both within and between global CDR categories were measured.

## 2.5 Software packages and code availability

The SillyPutty algorithm was implemented using the R package published in the Comprehensive R Archive Network (CRAN) https://cran.r-project.org/web/packages/SillyPutty/index.html. All other visualizations were performed using the seaborn package and Python 3.7. The implementation code for this project will be made available upon acceptance. The data used in our experiments is presented as a deanonymized table with no Protected Health Information (PHI) as S4 Table.

## 3. Results

### 3.1 Sample characteristics

Longitudinal data from 1,845 patients with 2,737 visits were eligible for inclusion, where each visit recorded a global CDR score. While 953 patients recorded a single visit, the remaining 892 patients had multiple visits (maximum number of visits = 5) with a visit interval of (mean +/- STD) 8.8 +/- 3.6 months. The median age of the cohort at the baseline visit of was 73 years with 57% of the patients being female. In terms of race and ethnicity, 77% patients were White and 9.8% patients were Black or African American. 88.5% patients were Non-Hispanic or Latino, while 5.1% were Hispanic or Latino (Table 1). As identified by codes from the 10th revision of the International Classification of Diseases (ICD-10), which are standardized alphanumeric codes maintained by the World Health Organization to allow consistent communication globally on specific diagnoses or health issues, the most frequently occurring brain-related disorders in our cohort included memory loss (N = 987 patients; 54%), Alzheimer's Disease (N = 1061 patients; 57.5%), Parkinsonism (N = 66 patients; 3.6%), Major Depressive Disorder (N = 455 patients; 24.7%) and Obstructive Sleep Apnea (N = 366 patients; 19.8%) respectively

**Table 1. Cohort demographics based on the baseline visits of patients.**

| Variable | Total |
|---|---|
| Number of patients | 1845 |
| Age at first encounter, median (IQR), years | 73 [64–81] |
| Sex, N (%) | |
| • Female | 1038 (57%) |
| Race, N (%) | |
| • White | 1422 (77%) |
| • Black or African American | 181 (9.8%) |
| • Asian | 20 (1.2%) |
| • *Other | 222 (12%) |
| Ethnicity, N (%) | |
| • Non-Hispanic or Latino | 1643 (88.5%) |
| • Hispanic or Latino | 84 (5.1%) |
| • Unknown | 118 (6.4%) |
| Brain-related disorders (n, %) | |
| • Memory Loss | 987 (54%) |
| • Alzheimer Disease | 1061 (57.5%) |
| • Parkinsonism | 66 (3.6%) |
| • Major Depressive Disorder | 455 (24.7%) |
| • Obstructive Sleep Apnea | 366 (19.8%) |

* Other includes Native Hawaiian or Other Pacific Islander, Other, Unknown, Declined, or unreported.

(Table 1). The full list of ICD-10 codes for each of the brain-related disorders listed in Table 1 can be found in the S2 Table.

## 3.2 Optimal set of features for clustering

We aimed to qualitatively analyze the T-SNE distribution of the 2 feature sets (CDR components and cognitive assessment scores; see Section 2.2.1) to decide which of them are the most optimal set of input features for clustering. Using the cognitive scores as features creates a gradient from low to high global CDR but did not result in clusters (Fig 2A). However, using the CDR components as features created clusters that were distinct without any significant overlap across the CDR categories (Fig 2B). Analysing the degree of overlap of points across the different global CDR categories, the six components of CDR were selected as the optimal set of features for clustering. Another motivation behind selecting the individual CDR component as features is the fact that for patients having the same global CDR score, the individual components might be different from one another, allowing us a more granular approach of studying the sub-phenotyping of patients.

Additionally, we conducted Principal Component Analysis (PCA) on both the cognitive assessment scores and the CDR components. Our results show that the first 2 principal components of the CDR components explain a significantly larger proportion of the variance (81.6% and 12.7% respectively) compared to the cognitive assessment scores (75.8% and 14.5% respectively). This quantitatively supports our decision to use CDR components as the primary features for clustering.

## 3.3 Clustering evaluation

**3.3.1 Optimal number of clusters.** Comparing the mean silhouette width (MSW) values for different number of clusters we identified K = 4, 6, 10, 15 as potential candidates where we

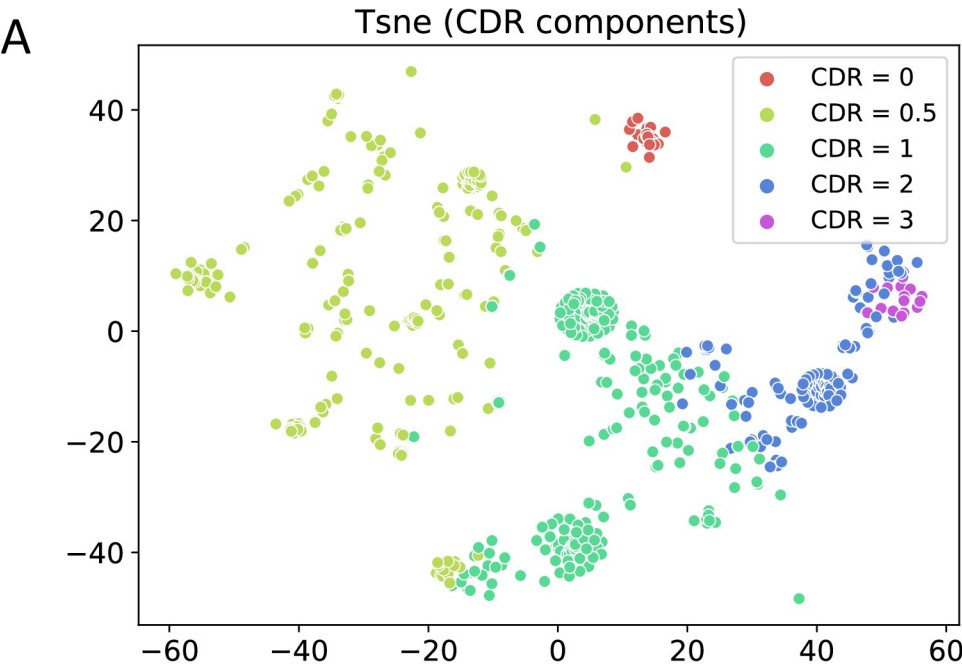

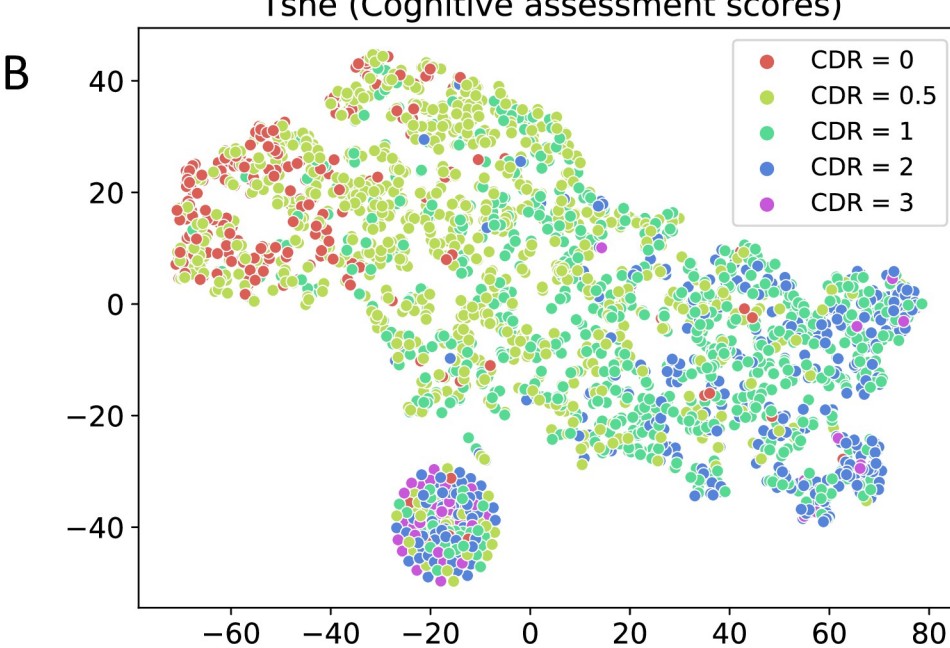

**Fig 2. Feature selection for clustering.** 2D T-SNE representations of the data for each of the 2 feature categories: CDR components only as features (**2A**) and cognitive scores only as features (**2B**). Each point in the scatter plot represents a visit with the colour indicating the global CDR score. The x-axis and y-axis represent the 2 dimensions of the 2D T-SNE vector for visualization purposes.

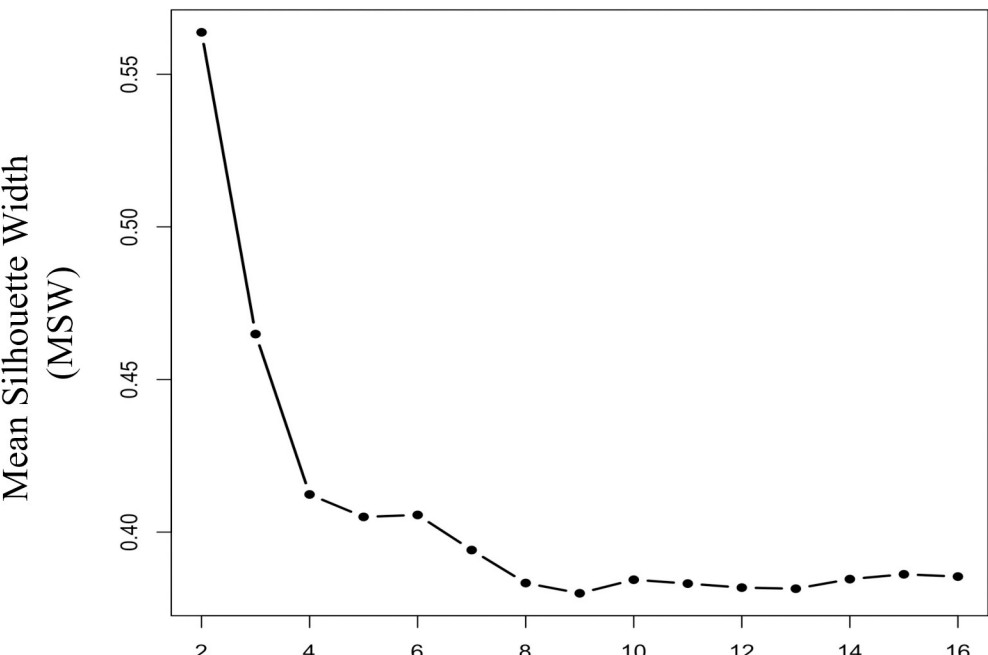

**Fig 3. Selecting number of clusters for hierarchical SillyPutty.** Optimal number of clusters were determined by comparing the best mean silhouette width (MSW), for number of clusters (K) ranging from 2 to 16. K = 10 was selected as the optimal number of clusters for our analyses (explanation provided in section 3.3.1).

observed increase in MSW compared to their neighbouring K value points (Fig 3). This aligns with T-SNE distributions (Fig 2B, S1 Fig), showing that higher number of clusters (K = 10, 15) better capture heterogeneity in cognitive profiles amongst early dementia patients (CDR = 0.5) compared to lower values of K (K = 4, 6). Choosing a lower value of K merges several smaller subclusters, which limits the ability to explore heterogeneity in greater detail. As K = 10 and K = 15 produced similar results, with K = 15 only further subdividing clusters for CDR > 1, we selected K = 10 as the optimal number of clusters, as our primary focus was to examine heterogeneity in the early stages of dementia (CDR ≤ 1).

**3.3.2 Comparison with other clustering methods.** Our method hierarchical SillyPutty was compared with the baseline clustering techniques using well-validated clustering metrics reported in the original SillyPutty paper. SillyPutty initialized with cluster assignments generated by hierarchical clustering showed higher mean silhouette width (MSW = 0.35715) across clusters compared to SillyPutty initialized with random cluster assignments (Random SillyPutty; MSW = 0.19989) and other state-of-the-art clustering algorithms like PAM (MSW = 0.3391) and hierarchical clustering (MSW = 0.30936) (Fig 4). Hierarchical SillyPutty also showed higher MSW compared to commonly used clustering methods such as K-means (MSW = 0.2678) and DBSCAN (MSW = 0.1985). Random SillyPutty performed the worst with 9 out of 10 clusters having negative silhouette widths indicating the possibility of misclassification of points to the wrong cluster or overlap between the clusters. Using only hierarchical clustering demonstrated negative silhouette widths in 7 out of 10 clusters. However, using the SillyPutty algorithm in combination with hierarchical clustering (HSP) showed positive silhouette widths for all the 10 clusters. The T-SNE and MDS plots for hierarchical SillyPutty showed more distinct and well-separated clusters compared to Random SillyPutty, PAM and hierarchical clustering (Fig 4).

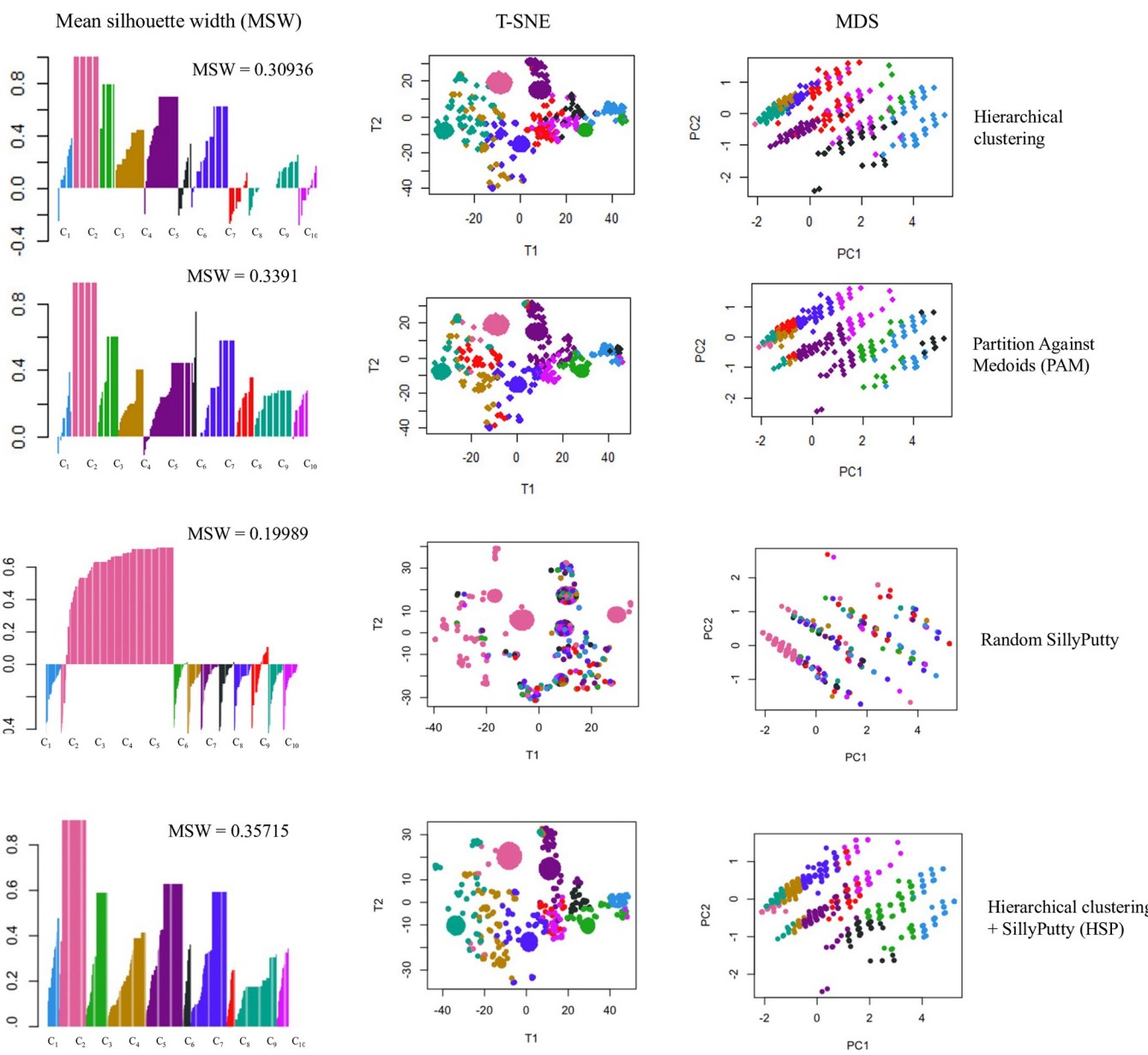

**Fig 4. Comparison with baseline clustering techniques.** Clustering statistics of different clustering algorithms: hierarchical clustering, PAM, random SillyPutty and our method hierarchical SillyPutty (from top to bottom). For each clustering method, the plots from left to right represent mean silhouette width (MSW), T-SNE distribution and multidimensional scaling (MDS) respectively. The clusters are arranged in ascending order ($C_1$- $C_{10}$) from left to right.

## 3.4 Subtype demographics

For each subtype, which were defined based on CDR component scores rather than global CDR, we aimed to examine if all patient visits within a subtype were in the same stages of dementia (unique global CDR score). Subtypes (clusters) can either be homogenous, having a unique global CDR (e.g. $C_9$, $C_4$), or composite, including two different global CDR scores, (e.g. $C_7$, $C_5$) (Fig 5). $C_2$ was a predominantly healthy cluster with patient visits associated with no cognitive impairment. $C_4$ and $C_9$ only consisted of visits with very mild dementia, while $C_7$,

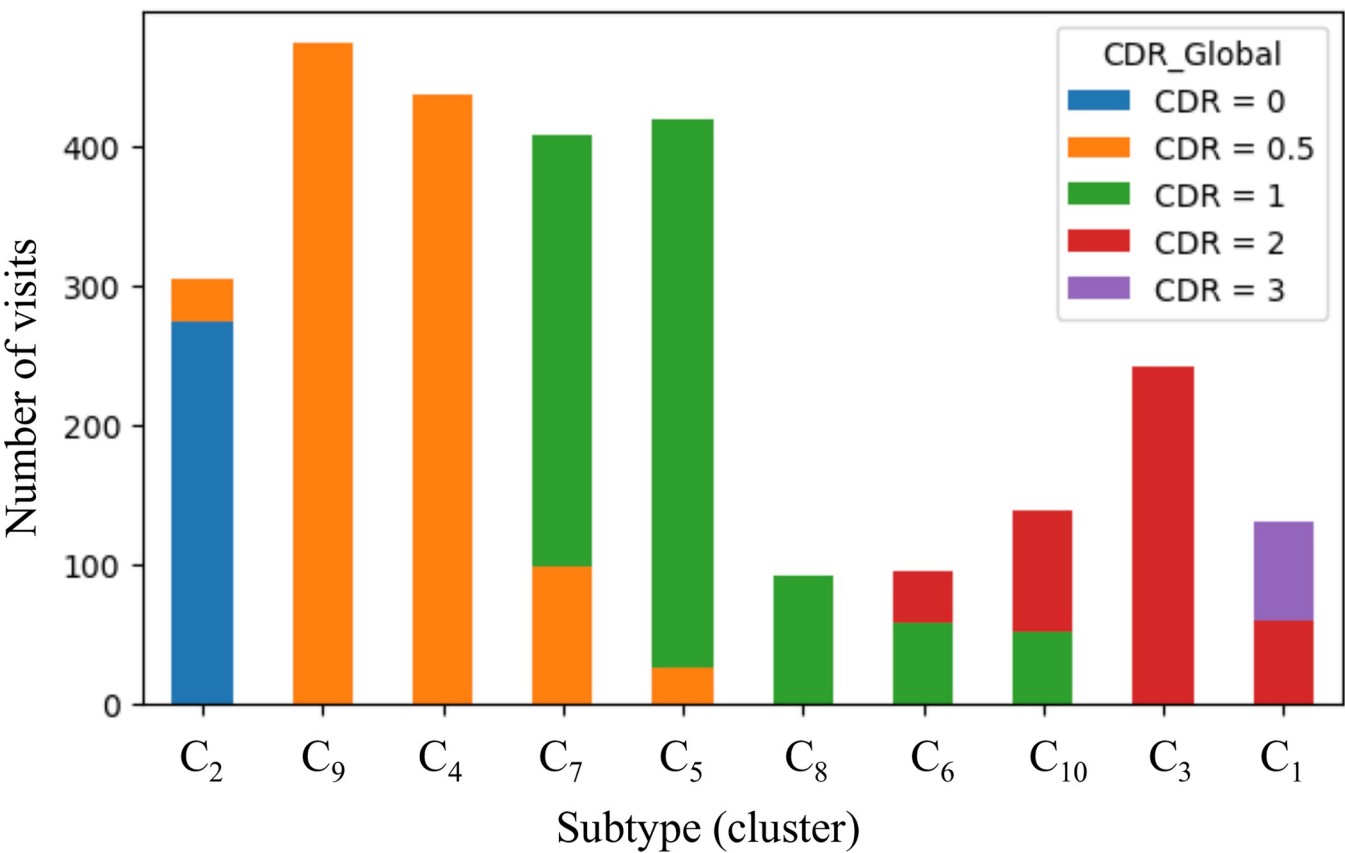

**Fig 5. Global CDR composition of subtypes.** Stacked bar plot showing the CDR composition of each dementia subtypes ordered by increasing global CDR score (dementia severity increases moving from left to right). The x-axis shows the 10 dementia subtypes, and the y-axis represents the number of visits within each dementia subtypes. Some dementia subtypes included a unique global CDR (e.g. $C_9$, $C_4$), while other dementia subtypes included two CDR scores (e.g. $C_7$, $C_5$).

$C_5$, $C_8$ and $C_6$ are mostly dominated by visits with mild dementia. Patient visits at a more advanced stages of the disease were mostly concentrated in clusters $C_{10}$, $C_3$, and $C_1$ respectively. More subtypes were associated with the mild dementia (CDR $<=1$), namely $C_2$, $C_9$, $C_4$, $C_7$, $C_5$, $C_8$ compared to moderate to severe dementia (CDR $> 1$) namely $C_6$, $C_{10}$, $C_3$, $C_1$. This is in line with the T-SNE distributions in Fig 2B and S3 Fig, where data points with CDR = 0.5 and 1 are more scattered, potentially forming multiple subtypes.

Demographic characteristics of patients (based on baseline visits) within each subtype are presented in Table 2. Patents with no cognitive impairment ($C_2$) were generally younger than the patients in other subtypes with mild to severe dementia. Patients across all 10 subtypes were mostly white and non-Hispanic (Table 2). Subtypes with very mld or mild dementia ($C_4$, $C_9$, $C_7$, $C_5$, $C_8$ and $C_6$) included a higher proportion of patients with memory issues, depression and sleep disorders compared to the subtypes with moderate or severe dementia (Table 2). Dementia patients diagnosed with AD were mostly assigned to the subtypes in the later stages of dementia ($C_{10}$, $C_3$, and $C_1$). For all the six cognitive assessment scores: Boston Naming Test, Mini-Mental State Exam, Short Blessed, Word List Memory Task, Logical Memory and Verbal Fluency, we observed a general trend of increasing cognitive impairment from early (mild) to later (moderate to severe) dementia stages, but not much variability across subtypes with same global CDR score (e.g. $C_4$ and $C_9$) (Table 2).

**Table 2. Cluster demographics based on baseline visits of each patient.** Similar to Fig 5, the clusters are arranged in the order of increasing dementia severity from left to right. * Other includes Native Hawaiian or Other Pacific Islander, Other, Unknown, Declined, or unreported.

| Variable | $C_2$ | $C_9$ | $C_4$ | $C_7$ | $C_5$ | $C_8$ | $C_6$ | $C_{10}$ | $C_3$ | $C_1$ |
|---|---|---|---|---|---|---|---|---|---|---|
| **Global CDR** | **0, 0.5** | **0.5** | **0.5** | **0.5, 1** | **0.5, 1** | **1** | **1, 2** | **1, 2** | **2** | **2, 3** |
| Number of unique patients | 194 | 388 | 281 | 293 | 301 | 47 | 51 | 73 | 158 | 59 |
| Age, median (IQR) | 69 (13) | 73 (12) | 76 (11) | 76 (13) | 75(12) | 77 (11) | 77 (15) | 77 (11) | 76 (12) | 72 (14) |
| Sex, (%) | | | | | | | | | | |
| • Female | 54.3% | 64.2% | 52.6% | 62.7% | 48.4% | 51.2% | 49.8% | 65% | 63.6% | 72.5% |
| Race, (%) | | | | | | | | | | |
| • White | 78.4% | 81.5% | 77.6% | 79% | 80.9% | 74.3% | 80.8% | 73.8% | 79.2% | 72.6% |
| • Black | 8.5% | 7.7% | 9.4% | 9.7% | 8.5% | 15.7% | 6% | 15.4% | 10.8% | 12.5% |
| • Asian | 0.82% | 1.2% | 1.3% | 0.9% | 0.82% | 0.93% | 0% | 0.85% | 0.82% | 0.98% |
| • *Other | 12.3% | 10.6% | 11.7% | 10.4% | 9.7% | 9.07% | 13.2% | 10% | 9.2% | 13.9% |
| Ethnicity, (%) | | | | | | | | | | |
| • Non-Hispanic | 92.2% | 89.5% | 91.3% | 88.6% | 93.1% | 87.4% | 91% | 87.5% | 91.4% | 91.6% |
| • Hispanic | 5.6% | 6.7% | 8.5% | 7.2% | 3.8% | 8.7% | 8.5% | 8.6% | 7.8% | 8.9% |
| Brain-related disorders, (%) | | | | | | | | | | |
| • Memory Loss | 86.5% | 64.2% | 57.6% | 52.3% | 57.5% | 75% | 58.8% | 38.2% | 29.7% | 31.5% |
| • Alzheimer Disease | 6.4% | 19.2% | 24.5% | 44.6% | 40% | 25% | 38.2% | 54.2% | 67.2% | 63.5% |
| • Parkinsonism | 0.46% | 1.3% | 1.36% | 1.2% | 1.6% | 0% | 30.6% | 0.8% | 2.1% | 1.4% |
| • Depression | 3.2% | 1.4% | 0.81% | 0.62% | 1.25% | 0% | 4.1% | 0% | 0% | 0% |
| • Sleep disorders | 3.4% | 2.9% | 1.9% | 1.2% | 0% | 0% | 1.1% | 1% | 1.4% | 4.5% |
| Cognitive assessment scores, median (IQR) | | | | | | | | | | |
| • Boston Naming Test | 15 (1) | 14 (2) | 14 (4) | 13 (3) | 13 (4) | 14 (4) | 13 (4) | 11 (6) | 10 (5) | 10 (3.5) |
| • MMSE | 28 (15) | 25 (14) | 22 (14) | 22 (13) | 19 (13) | 20 (13) | 18 (12) | 15 (11) | 16 (14) | 15 (9) |
| • Short Blessed | 2 (4) | 6 (9) | 12 (11) | 12 (10) | 18 (13) | 16 (14) | 17 (11) | 21 (8) | 21 (16) | 20 (8) |
| • Word List Memory | 18 (7) | 13 (5) | 12 (6) | 12 (4) | 11 (6) | 12 (5) | 9 (5.7) | 10 (7) | 10 (7) | 9 (5) |
| • Verbal Fluency | 15 (7) | 12 (6) | 11 (6) | 11 (5) | 9 (6) | 9 (6) | 8 (6) | 8 (7) | 9 (7) | 8 (4) |

## 3.5 Cognitive characteristics of identified subtypes

The association between the six CDR components and dementia subtypes allows for interpretation the cognitive profiles of these subtypes (Fig 6). The significance of these component scores lies in their ability to provide a detailed picture of the multifaceted nature of cognitive decline in dementia. For all the six CDR components, there is a natural progression of increasing cognitive impairment from early (mild) to later (moderate to severe) dementia stages. In our analyses, we analyzed the inter-subtype variability to examine if subtypes with the same CDR are heterogeneous in their CDR component scores (Fig 6). Further we also defined the cognitive profile of each subtype based on the distribution of each component score of patients within that subtype (Table 3).

Subtype $C_2$ consisted of a relatively healthy population with minimal cognitive and functional impairments (Table 3). Patients in both $C_4$ and $C_9$ recorded the same global CDR score of 0.5 and exhibited similar trends of mild to moderate memory loss and fully capable of self-care (Fig 6). However, $C_4$ visits were associated with slight disorientation, and slight impairment in judgement, community and home activities compared to patients in $C_9$ who were fully functional and healthy in those domains (Table 3). Comparing the subtypes ($C_7$, $C_5$, $C_8$ and $C_6$) with visits predominantly characterized by mild dementia (CDR = 1), we observed that the four subtypes showed similar level of moderate memory loss and orientation problems. However, patient visits in $C_8$ and $C_6$ showed more severe impairment in judgement &

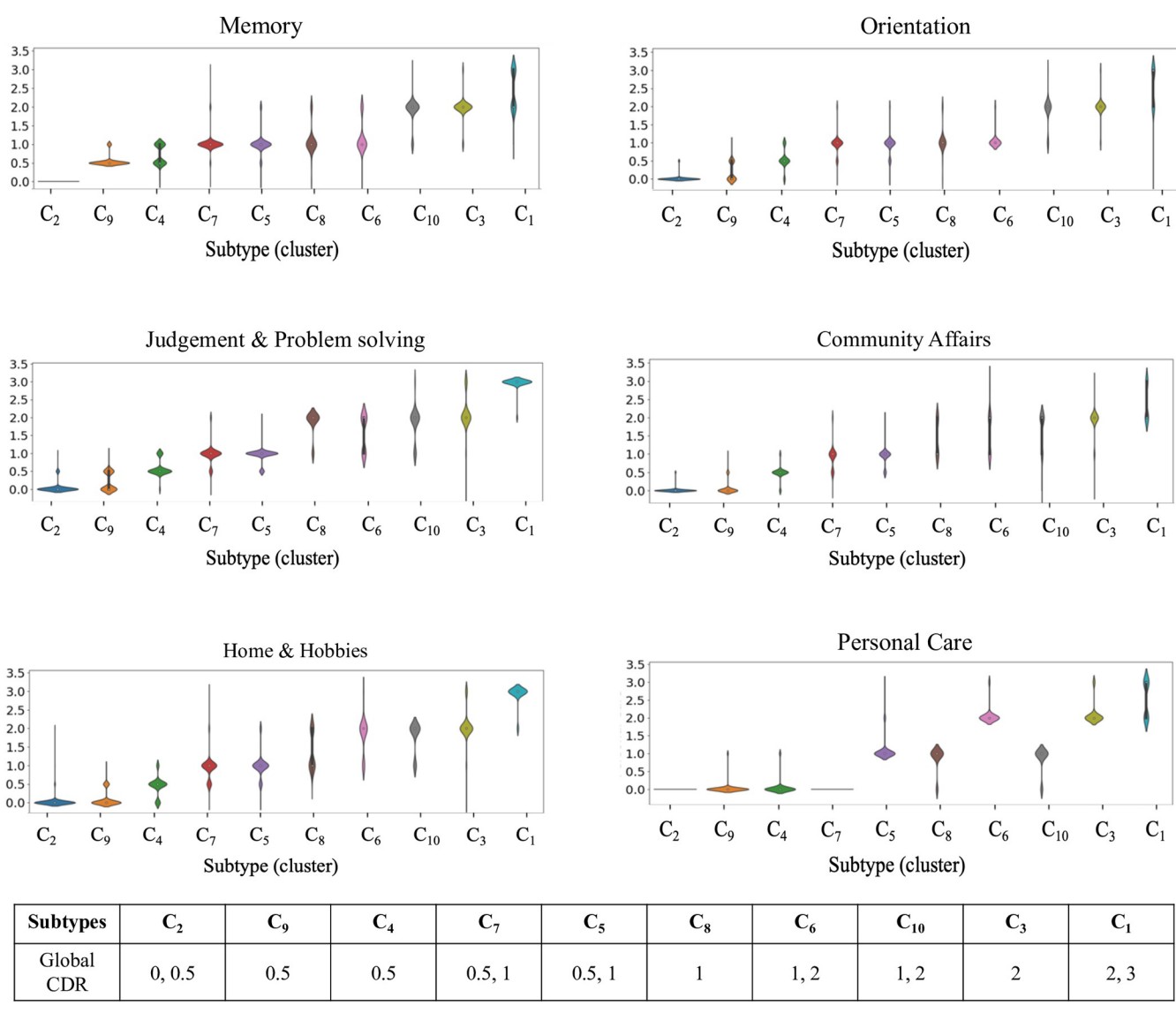

| Subtypes | $C_2$ | $C_9$ | $C_4$ | $C_7$ | $C_5$ | $C_8$ | $C_6$ | $C_{10}$ | $C_3$ | $C_1$ |
|---|---|---|---|---|---|---|---|---|---|---|
| Global CDR | 0, 0.5 | 0.5 | 0.5 | 0.5, 1 | 0.5, 1 | 1 | 1, 2 | 1, 2 | 2 | 2, 3 |

**Fig 6. Cognitive characteristics of subtypes.** Violin plots showing how each of the how each of the 6 components of CDR score vary across the 16 dementia subtypes. The x-axis represents the individual dementia subtypes. The y-axis shows the six CDR component scores varying from 0–3, with 0 indicating normal cognition and 3 indicating severe impairment. Similar to Fig 5, the clusters are arranged in the order of increasing dementia severity from left to right.

problem-solving ability, along with home and community activities compared to visits in $C_7$ and $C_5$ (Fig 6, Table 3). We observed considerable variability in self-care abilities between $C_7$, $C_5$, $C_8$ and $C_6$ respectively. While $C_7$ patients were fully capable of self-care, those in $C_5$, $C_8$ and $C_6$ required assistance in daily personal activities. The subtypes ($C_{10}$, $C_3$, and $C_1$) with visits predominantly characterized by moderate to severe dementia (CDR >1) showed similar extent of severe cognitive impairments in all the domains except personal care. While $C_{10}$ patients only needed prompting for daily activities, patients in $C_3$, and $C_1$ required significant assistance in dressing, hygiene and keeping of personal effects.

Overall, it can be concluded that subtypes at the early stages of dementia demonstrated more heterogeneity in cognitive characteristics compared to the ones at the later stages of the disease. The CDR components scores appear to be more sensitive to the heterogeneity in the cognitive characteristics compared to other cognitive assessments.

**Table 3. Cognitive profile summary (CDR components) of each subtype.** Subtypes have been grouped based on similarity in their cognitive characteristics. Similar to Fig 5, the clusters are arranged in the order of increasing dementia severity from top to bottom.

| Subtype | Global CDR | Cognitive profile summary |
|---|---|---|
| $C_2$ | 0, 0.5 | Relatively healthy population. Minimal cognitive and functional impairment. |
| $C_9$ | 0.5 | Mild to moderate memory loss, unimpaired functional capacity in other cognitive and functional domains. |
| $C_4$ | 0.5 | Mild to moderate memory loss. Slight disorientation, and slight impairment in judgement, community, and home activities |
| $C_7$ | 0.5, 1 | Moderate impairment in all six domains. Patients in $C_7$ were fully capable of self-care activities. |
| $C_5$ |  |  |
| $C_8$ | 1 | Moderate memory and orientation problems. Severe impairment in judgement, community, and home activities. |
| $C_6$ | 1, 2 |  |
| $C_{10}$ | 1, 2 | Severe impairment in all 6 cognitive and functional domains with maximum impairment level in $C_1$. $C_{10}$ showed moderate difficulty in personal care with need for caregiving support. |
| $C_3$ | 2 |  |
| $C_1$ | 2, 3 |  |

## 3.6 Cluster transitions

Finally, we examined how patients transitioned between the different subtypes over time and how these transitions were related to disease progression from CDR $< = 0.5$ to CDR $= 1$ (Fig 7A) and from CDR $= 1$ to CDR $>1$ (Fig 7B). With clustering performed on all visits of all patients in our cohort as opposed to only the baseline visit, we assumed that at any given point in time, a patient exists in a single subtype but transitions to a different subtype in their next visit. Out of 892 patients having multiple visits, 39% of them (n = 347) had subtype transitions corresponding to progression to more severe stages of dementia. We acknowledge that patients with a single visit cannot contribute to the analysis of transitions between stages and hence were excluded to measure transitions.

Our goal was to assess if subtypes at similar stages of dementia have variability in the risk of progression of advanced dementia stages. For example, patient visits in both $C_9$ and $C_4$ were characterized by very mild dementia (global CDR = 0.5), but patients in $C_4$ had more transitions to visits with mild dementia (global CDR = 1; $C_5$ $C_6$ and $C_7$), indicating greater chances of disease progression for $C_4$ visits (Fig 7A). On the contrary, majority of the transitions from $C_9$ were either within the same subtype (self-transitions) or to subtype $C_4$ at similar disease stages (CDR = 0.5). We also analyzed the median visit interval (time between successive visits in months) corresponding to $C_4$ and $C_9$ transitions to mild dementia (CDR = 1). We observed that the visit intervals were similar (p = 0.086) between $C_4$ (median = 9.2 months) and $C_9$ (median = 8.9 months). The above observations indicate the variability in the risk of progression for patient visits in 2 different subtypes at similar stages of dementia. The variability in the transitions to advanced dementia stages between $C_4$ and $C_9$ can be further linked to their heterogenous cognitive profiles, with $C_4$ patients having higher impairment in functional capacity compared to patients in $C_9$.

Analyzing the subtype transitions from mild dementia (global CDR = 1) to moderate/severe dementia (global CDR $> 1$), we observed variability in the chances of progression amongst subtypes with predominantly CDR = 1 visits ($C_5$, $C_6$, $C_7$ and $C_8$). $C_5$ and $C_7$ have more transitions to subtypes ($C_1$, $C_3$ and $C_{10}$) with moderate to severe dementia (CDR $> 1$) compared to $C_6$ and $C_8$ (Fig 7B). The median visit intervals for $C_5$, $C_7$, $C_6$ and $C_8$ transitions from mild to moderate/severe dementia were 7.5, 7.8, 8.1 and 7.9 months respectively. The number of pairwise transitions between all 10 subtypes are presented in the S3 Table.

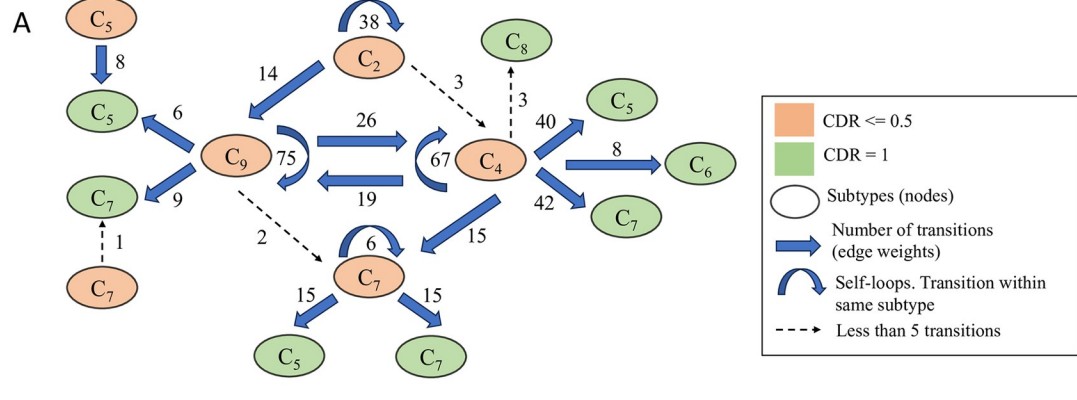

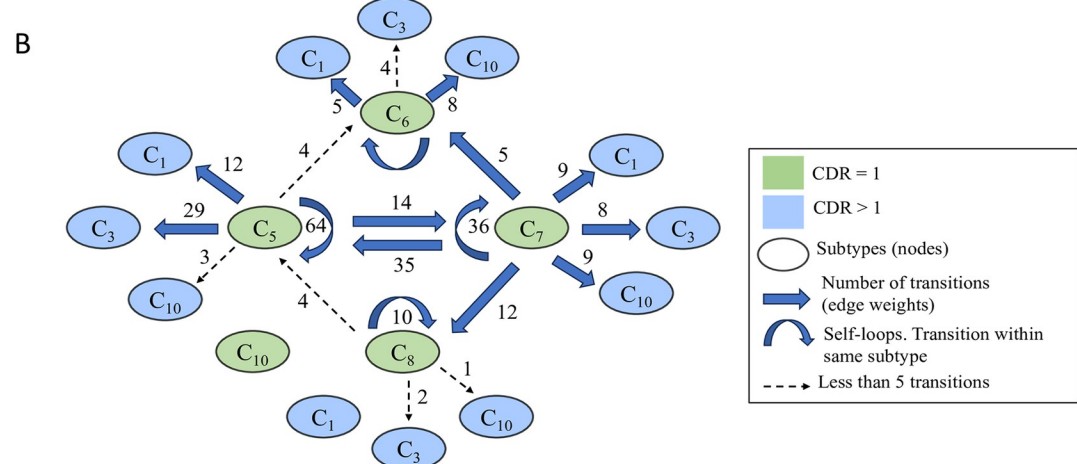

| Subtypes | $C_2$ | $C_9$ | $C_4$ | $C_7$ | $C_5$ | $C_8$ | $C_6$ | $C_{10}$ | $C_3$ | $C_1$ |
|---|---|---|---|---|---|---|---|---|---|---|
| Global CDR | 0, 0.5 | 0.5 | 0.5 | 0.5, 1 | 0.5, 1 | 1 | 1, 2 | 1, 2 | 2 | 2, 3 |

**Fig 7. Clusters transitions related to disease progression.** Graph showing transitions between subtypes which are related to global CDR progression from CDR < = 0.5 to CDR = 1 (**A**) and CDR = 1 to CDR > 1 (**B**). The nodes represent the subtypes, and the edge weights represent the number of transitions from source to target. Transitions with edge weights > = 5 are shown with dotted arrows. Transitions between nodes of same color including self-transitions indicate transitions remaining in the same CDR stage.

## 4. Discussion

In this study, we used the state-of-the-art unsupervised clustering method named SillyPutty, in combination with hierarchical clustering to estimate naturally occurring subtypes within a real-world clinical dementia cohort. We incorporated all longitudinal patient visits into our clustering analysis, instead of relying only on baseline visits, allowing us to explore the ongoing relationship between subtypes and disease progression over time. Our results demonstrated that subtypes with very mild or mild dementia exhibited more variability in their cognitive characteristics and risk of disease progression.

Neuroimaging biomarkers such as MRI and PET have been predominantly used for estimating subtypes to assess heterogeneity in AD. Compared to MRI, which requires specialized

equipment and facilities, cognitive assessment scores are more easily accessible and can be conducted in various settings, such as clinics and homes, making them suitable for large-scale screenings. The six component measures of CDR used in our analyses examined different cognitive and functional domains providing critical insights into different aspects of a patient's cognitive health due to dementia. These component measures are closely linked to functional abilities and daily living skills, which are critical for evaluating the impact of dementia on a patient's quality of life. Further, our results demonstrate that patients with the same CDR can have different cognitive characteristics with respect to the different CDR components. We believe that using the CDR component scores as clustering features is a cost-effective promising approach to investigate cognitive heterogeneity in dementia, which can be helpful effective clinical decision-making and precision diagnostics tailored to each subtype.

Results showed that both subtypes $C_4$ and $C_9$ were at early stages of dementia with mild cognitive impairment (MCI; global CDR = 0.5), but had different cognitive profiles and risk of progression. For example, higher number of transitions to advanced dementia stages for $C_4$ compared to $C_9$ indicated that patients in $C_4$ and $C_9$ can be considered as progressive MCI and stable MCI respectively. Further, $C_4$ patients have more impairment in orientation, judgement & problem solving, home & hobbies and community affairs compared to patients in $C_9$. These particular cognitive and functional domains might be useful features to distinguish between patients with stable or progressive cognitive decline. Our results also indicated that subtypes with visits having mild dementia were more heterogenous in their cognitive profiles compared to subtypes with visits having moderate to severe dementia. Identifying and diagnosing individuals in the initial stages of their cognitive decline early can enable timely intervention, which might slow the progression of the disease and improve the quality of life. For instance, patients identified as being at higher risk for rapid cognitive decline (e.g. C4) could benefit from more aggressive therapeutic interventions (e.g., early pharmacological treatment, lifestyle modifications) or more frequent cognitive monitoring. Additionally, this stratification could help prioritize patients for clinical trials targeting slowing cognitive decline, thereby optimizing resource allocation and improving patient outcomes.

In our analyses, we focused primarily on the more common case of forward transitions toward increasingly severe impairments (mild to severe dementia), as dementia is generally a progressive and irreversible condition. Reverse transitions (from more severe to milder clusters) on the other hand were rarely observed (S3 Table). The regression in CDR score could be potentially related to treatment of conditions that can cause or worsen dementia symptoms. For example, dementia specialists often discontinue medications that impair cognition, and patients often improve after these changes are made [39, 40].

In this study, the unsupervised clustering analysis was performed using all aggregated visits of every patient, instead of using only their baseline visits. This design choice was motivated by the following reasons. First, analyzing visit-level data enables the examination of temporal patterns and transitions between dementia subtypes over time. This helps in understanding how dementia progresses in different patients and how they may transition between various stages and subtypes, providing insights into the disease's trajectory that are not possible with a baseline-only approach. Second, considering each visit as a separate data point significantly increases the volume of data available for analysis. This is particularly important when the patient cohort is small, as it enhances the statistical power and reliability of the clustering results. Finally, dementia often progresses over many years, but our electronic health record (EHR) data spans only five years (2013–2018). Comprehensive longitudinal analysis is possible only if all visits for each patient are available. However, our dataset included patients with different windows of their clinical care (e.g. some may have been seen in 2006–2014, others from 2016–2022). With the availability of EHR data only from 2012–2018, we chose to cluster at the

visit level. While patients with a single visit do not contribute to longitudinal transitions, they were still included in the clustering process to maximize the sample size and enhance the statistical power of the clustering algorithm. Including these single-visit patients helped to more accurately define cluster characteristics and ensured that our model captured the full spectrum of the cohort's heterogeneity.

Our study has some important limitations worth considering. First, our pipeline is built on a single EHR dataset. As part of future work, we plan to test the generalizability of our pipeline on additional datasets. Second, the sample size of our dataset (dementia patients with multiple visits who made a transition) is relatively small which restricts our ability to perform rigorous statistical analyses on the variable rate of progression of dementia subtypes. Applying this pipeline to a larger EHR dataset can allow us to estimate probabilistic estimates of disease progression, backed by rigorous statistical analysis. Third, cognitive status at one visit may be influenced by prior visits, particularly in progressive conditions like dementia. While our current approach focuses on clustering independent data points, future research could explore more sophisticated longitudinal approaches, such as Hidden Markov Models (HMM), to better capture the temporal dynamics of cognitive decline and progression in dementia. Fourth, our experiments did not consider potential confounding factors such as comorbidities, medication status and socio-economic status (SES) which could potentially influence the progression risks of certain clusters despite similar initial profiles. Incorporating additional data on comorbidities, medication, and SES to better understand their impact on cluster progression can be an important avenue for further research. Finally, our analysis did not adopt a patient-centered perspective, focusing instead on aggregate data rather than individual trajectories. This presents an opportunity for future research to employ a different analytical approach, considering data from a more granular, patient-centric viewpoint.

## 5. Conclusions

In our study, we applied a recently developed data-driven unsupervised clustering algorithm named hierarchical SillyPutty on the CDR component scores and identified 10 subtypes within a real-world clinical dementia cohort. Unlike previous approaches, we included all longitudinal patient visits in our clustering analysis, which allowed us to investigate the dynamic relationship between subtypes and disease progression over time. Our findings indicated that subtypes with very mild or mild dementia exhibited greater heterogeneity in cognitive profiles and varied risks of disease progression.

## Supporting information

**S1 Fig. T-SNE distribution stratified by clusters.** T-SNE distribution of the clustering results for different values of K = 4, 6, 10 and 15 respectively. The colorbar represents the different clusters. Each point in the scatter plot represents a single visit. The x-axis and y-axis represent the 2 dimensions of the 2D T-SNE vector for visualization purposes.
(PDF)

**S1 Table. Detailed information about cognitive assessment scores and CDR components (section 2.2.1).**
(DOCX)

**S2 Table. ICD 10 and ICD 9 codes used to identify patients with the brain-related disorders presented in Tables 1 and 2.**
(DOCX)

**S3 Table. Number of transitions between every pair of dementia subtype groups.** The subtype groups in the columns and rows represent source and target respectively. The last column represents the total number of transitions from each source subtype. The diagonal values (marked in bold red) represent the number of self-transitions or instances where patients transition between the same subtype. Similar to Fig 5, the clusters are arranged in the order of increasing dementia severity from left to right.
(DOCX)

**S4 Table. Deanonymized data file used in our analyses with no Protected Health Information (PHI).** The ID column has been recoded from 1 to n where n is the number of patients. The data frame was sorted according to visit date and all visit dates were deleted from the table.
(CSV)

## Author Contributions

**Conceptualization:** Sayantan Kumar, Zachary Abrams, Philip R. O. Payne.

**Data curation:** Sayantan Kumar.

**Formal analysis:** Sayantan Kumar.

**Funding acquisition:** Philip R. O. Payne.

**Investigation:** Philip R. O. Payne.

**Methodology:** Sayantan Kumar, Zachary Abrams.

**Project administration:** Philip R. O. Payne.

**Software:** Sayantan Kumar, Zachary Abrams.

**Supervision:** Zachary Abrams, Philip R. O. Payne.

**Validation:** Sayantan Kumar, Zachary Abrams.

**Visualization:** Sayantan Kumar.

**Writing – original draft:** Sayantan Kumar.

**Writing – review & editing:** Sayantan Kumar, Inez Y. Oh, Suzanne E. Schindler, Nupur Ghoshal, Zachary Abrams, Philip R. O. Payne.

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
