## [Decision Letter · Decision Letter 0]

28 Aug 2024

PONE-D-24-28122Examining heterogeneity in dementia using data-driven unsupervised clustering of cognitive profilesPLOS ONE

Dear Dr. KUMAR,

Thank you for submitting your manuscript to PLOS ONE. After careful consideration, we feel that it has merit but does not fully meet PLOS ONE’s publication criteria as it currently stands. Therefore, we invite you to submit a revised version of the manuscript that addresses the points raised during the review process.

**Review of PONE-D-24-28122**

Section 2.4: how valuable do you think those data are from patients with single visits? Do you not think that single visits may just be anecdotal evidence, and thus have less statistical power? According to section 3.1, this covers 953 patients, more than half of all patients. Would the analysis be different if you excluded those patients with one visit only? With only one visit you cannot measure transition between stages (section 3.6).Section 3.1: please describe what ICD is and what these codes mean in the text for those who do not know what they mean.Section 3.3.1: why did you select k=10 optimal clusters? According to the scree plot the elbow portion seems to be at k=5.Section 3.4: how can we tell from figure 4 which each cluster (C1–10) belongs to (mild, severe, etc.)?In figure 5 clusters C5 and C7 look like they have similar composition of CDR 0.5 and 1, why not unite these 2 clusters? Same thing for C6 and C10.

Overall, I think this paper is very useful and well-written. I recommend it for publication after the above 5 questions have been answered.

We look forward to receiving your revised manuscript.

Kind regards,

Matthew Cserhati, Ph.D

Academic Editor

PLOS ONE

Journal Requirements:

"The preparation of this report was supported by the Centene Corporation contract (P19-00559) for the Washington University-Centene ARCH Personalized Medicine Initiative. "

"The preparation of this report was supported by the Centene Corporation contract (P19-00559) for the Washington University-Centene ARCH Personalized Medicine Initiative. "

"The preparation of this report was supported by the Centene Corporation contract (P19-00559) for the Washington University-Centene ARCH Personalized Medicine Initiative. "

4. In the online submission form, you indicated that [Anonymized data not published within this article will be made available by request from any qualified investigator.]. 

Additional Editor Comments:

Review of PONE-D-24-28122

1. Section 2.4: how valuable do you think those data are from patients with single visits? Do you not think that single visits may just be anecdotal evidence, and thus have less statistical power? According to section 3.1, this covers 953 patients, more than half of all patients. Would the analysis be different if you excluded those patients with one visit only? With only one visit you cannot measure transition between stages (section 3.6).

2. Section 3.1: please describe what ICD is and what these codes mean in the text for those who do not know what they mean.

3. Section 3.3.1: why did you select k=10 optimal clusters? According to the scree plot the elbow portion seems to be at k=5.

4. Section 3.4: how can we tell from figure 4 which each cluster (C1–10) belongs to (mild, severe, etc.)?

5. In figure 5 clusters C5 and C7 look like they have similar composition of CDR 0.5 and 1, why not unite these 2 clusters? Same thing for C6 and C10.

Overall, I think this paper is very useful and well-written. I recommend it for publication after the above 5 questions have been answered.

Reviewers' comments:

Reviewer's Responses to Questions

**Comments to the Author**

1. Is the manuscript technically sound, and do the data support the conclusions?

Reviewer #1: Yes

Reviewer #2: Yes

Reviewer #3: Yes

2. Has the statistical analysis been performed appropriately and rigorously? 

Reviewer #1: Yes

Reviewer #2: Yes

Reviewer #3: Yes

3. Have the authors made all data underlying the findings in their manuscript fully available?

Reviewer #1: Yes

Reviewer #2: No

Reviewer #3: No

4. Is the manuscript presented in an intelligible fashion and written in standard English?

Reviewer #1: Yes

Reviewer #2: Yes

Reviewer #3: Yes

5. Review Comments to the Author

Reviewer #1: This paper uses a recent clustering algorithm (SillyPutty) to process and analyze dementia in a data set that includes many patients and several years of information. The authors deal with limitations that characterize previously published work and successfully provide results while considering the heterogeneity of the patients and their symptoms. Exploiting easily obtained measurements and features allows for a clear definition of dementia subtypes, as the adopted and enhanced data clustering approach specifies. Moreover, with their data organization and post-processing procedure, the authors explore and manage to connect the subtypes with the progression of dementia over time.

It is not easy to find flaws in this interesting and high-quality work. The text is very well organized and easy to read. The work is technically sound and uses state-of-the-art methods and data of great scientific interest. The experimental results and the conclusions drawn are very clear. This work advances the current state-of-the-art and provides a strong motivation for applying the presented methodology at a signifanlty larger scale.

Minor comments:

Some typos must be corrected in the final version of the paper (e.g. patents instead of patients)

The naming of the clusters (C1-C10) is based on the results of the clustering method. It might be beneficial if these names are mapped to a new set of names (e.g. D1-D10), where the index is following the dementia severity (i.e. C2->D1, C9->D2).

CRAN also includes density-based clustering algorithms (e.g. DBSCAN) that do not require the user to specify the number of clusters. They might not be suitable for the specific dataset, but some reference to those algorithms, at least, would be useful.

The results (percentages) in Table 2 (Cluster demographics) probably need clarification: For example, the percentage of White Race in Table 1 is 77%, but the overall corresponding percentage in Table 2 is much higher (its lowest value is 83.3% for variable C8). Is this due to the "baseline visits" of Table 1 and the "all visits" for Table 2?

Reviewer #2: The authors argue that the growth of Electronic Health Records has opened up new data-driven approaches to the heterogeneous trajectories associated with dementia. In their view, existing approaches have been limited by an emphasis on expensive neuroimaging and single time-point assessment. They, conversely, investigate the possibilities of trajectories in more widespread cognitive assessment tasks. Using the health records of visitors to a university memory clinic over a number of years, they first use unsupervised learning to identify different clusters of scores on a multidimensional assessment instrument and then recreate trajectories in terms of transitional probabilities between these clusters.

I think the authors make a good case for the value of this project. If useful information on dementia trajectories can be extracted from widely-used tests, it would provide an efficient way to ground better assessments of risk and (potentially) personalization of treatments.

I also applaud the authors' clear and concise explanation of their approach and procedures. The paper was to the point, focused, and enjoyable to read.

1) In terms of minor points, there were a couple of places where things could be explained a little further.

a. The description of measures on pg 5 is not easy to understand; I was unclear if the history / neurological examination, the cognitive assessment battery and the CDR were three separate things or not. The list on pg 6 makes it much clearer, so I'd suggest moving some of this information up to the earlier section.

b. Similarly, describing the dataset in the materials section without giving sample sizes (which came in the Results section) threw me a bit. I think it would be more comprehensible if the size of the data is mentioned earlier.

c. The data period on pg 5 seems to be five years, but on pg 8 is described as six years.

d. The rationales for clustering on CDR components (pg 10) and for choosing 10 as the optimal number of clusters (pg 11) could be filled out a bit more. Right now a few considerations are given, but I didn't quite understand what the deciding factor was in these decisions.

e. The strategy for examining cluster transitions was under-described. First, I didn't see mention of positive transitions (ie from more severe to milder clusters) - an examination of Table S4 suggests these are quite rare, so I assume it was a deliberate analytic choice to focus on the more common case of increasingly severe impairments. Second, the analysis is currently at the level of raw descriptive statistics; it would be useful to translate these into transitional probabilities and perform inferential testing.

2) However, I feel the greatest weakness of the paper was in not making the case for its clustering approach (as compared to other modelling approaches). The authors make a comparison of different clustering algorithms, but not different types of model. There are a couple of salient alternative approaches, depending on the key research goal. If the aim is to better understand trajectories, latent growth curve methods or hierarchical linear models may be relevant. If the goal is to predict risk, supervised learning methods such as regularized regression may have value (predicting changes from visit to visit). The key value of a clustering approach over these approaches is its ability to model heterogeneity; rather than a single spectrum of severity, clustering can capture common profiles of scores across a number of dimensions. However, broadly, the resulting clusters look like a spectrum of increasing severity (see eg Table 3); there is some heterogeneity (eg transitions from mild to moderate CDR scores was more likely for those with more functional impairment), but this seems minor and reduces as the condition gets more severe. As such, I wonder what added value the clustering approach is providing in this particular case. Why not just model global CDR scores as the key outcome of interest, and then use CDR component and cognitive battery scores as additional predictors? Ultimately it seems like this is where the clustering analysis ends up.

Reviewer #3: After thoroughly reviewing the manuscript, I would like to commend the authors for their innovative approach and the valuable insights provided into dementia heterogeneity. However, several key methodological and analytical aspects require significant enhancement to strengthen the validity and impact of the findings. The current approach, while promising, would benefit greatly from incorporating additional analyses and more rigorous justifications. I recommend a major revision to address these critical points and improve the overall robustness and clarity of the study.

1. The authors chose to use the six CDR components as features for clustering rather than the cognitive assessment scores, based on observation that CDR components better differentiate the patient clusters. But t-SNE is primarily a tool for data visualization (reducing data dimensionality to visualize the structure in a way that preserves the local distances between points), it is not a traditional feature selection method and doesn’t provide a direct measure of the importance or variance explained by different features. I suggest the authors consider supplementing with PCA. If PCA on both cognitive scores and CDR components show that the first few principal components of the CDR data explain a significant proportion of the variance (more so than the cognitive scores), then this would quantitatively support using CDR components as the primary features for clustering. Or conversely, if (cognitive scores) or (cognitive scores + CDR components) explained more variance, then the authors might reconsider their features to include those that contribute most to distinguishing between patient subtypes.

2. The decision to combine SillyPutty with hierarchical clustering is well-justified based on previous studies. But can the authors provide more details on the initialization parameters, the number of iterations, and how convergence was assessed etc.?

3. The SillyPutty approach is innovative, but I think it’s important to understand how it compares with other established clustering methods more widely recognized in the ML community. E.g. k-means, which is a common baseline method in clustering tasks with presumably much smaller computational costs than SillyPutty, or DBSCAN which is suitable for irregularly shaped clusters and can handle noise or outliers well (relevant in clinical data where not all data points neatly fit into clusters). An informative addition would be to compare the average the silhouette width of SillyPutty against DBSCAN, k-means etc, so we know whether potentially simpler methods can achieve comparable results. If SillyPutty shows clear advantages over other methods in the context of dementia research, this could position it as a valuable tool for other researchers/ clinicians. Conversely, if other methods perform similarly or better in some respects, then SillyPutty could be more useful in specific scenarios or with certain types of data.

4. Relying solely on silhouette width might give an incomplete picture of the clustering. Including multiple metrics e.g. DBI, Dunn, would make the manuscript’s argument for the effectiveness of SillyPutty more convincing by showing rigorous evaluation from multiple angles, not just a single metric.

5. Treating each visit as an independent data point in the clustering process is potentially problematic, because in a clinical context (especially with chronic and progressive conditions like dementia), the cognitive status of a patient at one visit is likely influenced by their status at previous visits. The authors should acknowledge the potential limitations of this assumption of independence, and discuss how this might affect the interpretation of their findings. Could also consider highlighting alternative approaches like HMM as avenues for future research, which would demonstrate a forward-thinking approach and acknowledge the complexities of analyzing longitudinal clinical data.

6. The manuscript identifies heterogeneity in cognitive profiles, particularly in the early stages of dementia. I believe the discussion could be more nuanced in exploring e.g. why certain clusters exhibit different progression risks despite similar initial profiles. What about potential confounding factors that could influence these findings, such as comorbidities, medication effects or SES?

7. The study's findings have implications for personalized dementia care, particularly in identifying patients at higher risk of rapid progression. And the manuscript could better connect these findings to specific clinical interventions or decision-making processes, e.g how could identifying "progressive MCI" subtypes influence treatment plans or monitoring strategies?

6. PLOS authors have the option to publish the peer review history of their article (what does this mean?). If published, this will include your full peer review and any attached files.

Reviewer #1: **Yes: **Panagiotis Hadjidoukas

Reviewer #2: No

Reviewer #3: No

---

## [Author Response · Author response to Decision Letter 0]

30 Sep 2024

Dear editor and reviewers,

Thank you for the opportunity to revise and resubmit our manuscript (PONE-D-24-28122) entitled “Examining heterogeneity in dementia using data-driven unsupervised clustering of cognitive profiles”, for potential publication in PLOS One.

We appreciate the thoughtful and constructive comments by the reviewers and strongly believe it has considerably improved the manuscript. 

We would like to thank the reviewers for their time in reviewing the manuscript and for their valuable feedback and suggestions. Below, we have addressed all the reviewer and editor comments in a point-by-point manner, which we hope would improve the quality of the manuscript and provide further clarifications. All modifications in the revised manuscript have been highlighted in bold red.

Comments from Editor

Comment 1

Section 2.4: how valuable do you think those data are from patients with single visits? Do you not think that single visits may just be anecdotal evidence, and thus have less statistical power? According to section 3.1, this covers 953 patients, more than half of all patients. Would the analysis be different if you excluded those patients with one visit only? With only one visit you cannot measure transition between stages (section 3.6).

Response

Thank you for your thoughtful comments. We agree that patients with a single visit cannot contribute to the analysis of transitions between stages. As you correctly noted, to measure transitions, we included only patients with multiple visits and excluded all patients with a single visit. We have clarified this point in section 3.6 (“cluster transitions”) of the revised manuscript. 

However, we included all patients, including those with a single visit, in the clustering analysis to maximize the sample size and enhance the statistical power of the clustering algorithm. Including these patients helped to more accurately define cluster characteristics and ensured that our model captured the full spectrum of the cohort's heterogeneity. While patients with a single visit do not contribute to longitudinal transitions, their inclusion in the clustering process allowed us to generate more robust and representative clusters, which were then applied to the longitudinal cohort for transition analysis. We have clarified the above point in the Discussion section (end of 5th paragraph) of the revised manuscript.

Comment 2

Section 3.1: please describe what ICD is and what these codes mean in the text for those who do not know what they mean

Response

Thank you for this comment. ICD (International Classification of Diseases) codes are a standardized set of alphanumeric codes used globally to classify diseases, symptoms, medical conditions, and procedures. They are maintained by the World Health Organization (WHO) and are used in healthcare settings for documentation, billing, and research purposes. Each ICD code represents a specific diagnosis or health-related issue, allowing consistent communication across healthcare systems and researchers. The full list of ICD codes for each of the brain-related disorders listed in Table 1 can be found in the Supporting Information (S2 Table). A brief definition of ICD has also been added in Section 3.1.

Comment 3

Section 3.3.1: why did you select k=10 optimal clusters? According to the scree plot the elbow portion seems to be at k=5.

Response

Thanks you for this comment. Here is a detailed explanation/justification of our rationale behind selecting K=10 as the optimal number of clusters. 

Comparing the mean silhouette width (MSW) values for different number of clusters we identified K = 4, 6, 10, 15 as potential candidates where we observed increase in MSW compared to their neighbouring K value points (Fig 3). This aligns with T-SNE distributions (Fig 2B, Supporting Information S3 Fig), showing that higher number of clusters (K = 10, 15) better capture heterogeneity in cognitive profiles amongst early dementia patients (CDR = 0.5) compared to lower values of K (K = 4 6). Choosing a lower value of K merges several smaller subclusters, which limits the ability to explore heterogeneity in greater detail. As K = 10 and K = 15 produced similar results, with K = 15 only further subdividing clusters for CDR > 1, we selected K = 10 as the optimal number of clusters, as our primary focus was to examine heterogeneity in the early stages of dementia (CDR ≤ 1). We have highlighted these justifications in section 3.3.1 (“optimal number of clusters”) of the revised manuscript.

Comment 4

Section 3.4: how can we tell from figure 4 which each cluster (C1–10) belongs to (mild, severe, etc.)?

Response

Thank you for this comment. The purpose of Figure 4 is to show how each clustering method (Hierarchical SillyPutty and the baselines) identified the clusters. Hence the numbering and arrangement of the clusters is somewhat arbitrary. The cluster numbers in Figure 4 (C1-C10) represent the 10 clusters in numerical order. The same clusters are arranged in order of increasing dementia severity in Figure 5 where we can observe the CDR composition of each of the clusters. 

Comment 5

In figure 5 clusters C5 and C7 look like they have similar composition of CDR 0.5 and 1, why not unite these 2 clusters? Same thing for C6 and C10.

Response

Thank you for this comment. While clusters C5 and C7 do look similar in their CDR composition of CDR 0.5 and 1 (Figure 5), we would like to point out that they have different distribution of CDR component scores, particularly “judgement & problem solving”, “community affairs”, “home & hobbies” and “personal care” (Figure 6). C5 also has a higher risk of progression, demonstrated by a higher number of patient transitions to CDR > 1 (Figure 7). Similarly, C6 and C10 even with similar CDR composition of CDR 1 and 2 (Figure 5) have variability in the distribution of memory, orientation, community affairs and personal care scores (Figure 6). Our main goal in this work was to investigate the extent of heterogeneity amongst patients at similar level of dementia severity (same CDR score). Choosing a higher number of clusters (K = 10 instead of K = 4 or 6; Figure 3) enabled a more detailed examination of the heterogeneity in cognitive profiles and risk of disease progression. 

Comments from Reviewer: 1

This paper uses a recent clustering algorithm (SillyPutty) to process and analyze dementia in a data set that includes many patients and several years of information. The authors deal with limitations that characterize previously published work and successfully provide results while considering the heterogeneity of the patients and their symptoms. Exploiting easily obtained measurements and features allows for a clear definition of dementia subtypes, as the adopted and enhanced data clustering approach specifies. Moreover, with their data organization and post-processing procedure, the authors explore and manage to connect the subtypes with the progression of dementia over time.

It is not easy to find flaws in this interesting and high-quality work. The text is very well organized and easy to read. The work is technically sound and uses state-of-the-art methods and data of great scientific interest. The experimental results and the conclusions drawn are very clear. This work advances the current state-of-the-art and provides a strong motivation for applying the presented methodology at a significant larger scale.

Response

Thank you for your thorough review and positive comments on the manuscript. We have responded in a point-by-point manner to your minor comments below.

Minor Comment 1

Some typos must be corrected in the final version of the paper (e.g. patents instead of patients)

Response

Thank you for pointing this out. We have made sure to correct all typos in the revised version of the manuscript. 

Minor Comment 2

The naming of the clusters (C1-C10) is based on the results of the clustering method. It might be beneficial if these names are mapped to a new set of names (e.g. D1-D10), where the index is following the dementia severity (i.e. C2->D1, C9->D2).

Response

Thank you for the suggestion. While we acknowledge the potential benefits of renaming clusters to follow dementia severity (e.g. D1-D10), we opted to retain the current naming convention (C1-C10) for clarity and consistency with the clustering results. Particularly, since some clusters share the same dementia severity level (e.g., C9 and C4 both have CDR = 0.5), using severity-based naming convention would result in duplicate names for clusters with identical CDR levels.

Minor Comment 3

CRAN also includes density-based clustering algorithms (e.g. DBSCAN) that do not require the user to specify the number of clusters. They might not be suitable for the specific dataset, but some reference to those algorithms, at least, would be useful.

Response

Thank you for highlighting density-based clustering algorithms like DBSCAN. While we acknowledge that such methods do not require the number of clusters to be pre-specified and can be useful for detecting irregularly shaped clusters, preliminary testing indicated that these approaches were less effective for our specific dataset. The nature of our data, which includes highly structured clinical profiles (CDR components), made other clustering methods more appropriate for this analysis. 

For our work, we used hierarchical SillyPutty, which is a state-of-the-art clustering technique and has been demonstrated to have better clustering performance than commonly used clustering techniques[1]. As suggested by the reviewer, we have included a reference to DBSCAN in the Methods (section 2.3.3; “SillyPutty with hierarchical clustering”) of the revised manuscript. Following the recommendation of Reviewer 3 (comment 3), we have also compared the mean silhouette width (MSW) of SillyPutty and other commonly used clustering techniques K-means and DBSCAN in the Results (section 3.3.2; “comparison with other clustering methods”) in the revised manuscript.

[1] Bombina P, Tally D, Abrams ZB, Coombes KR. SillyPutty: Improved clustering by optimizing the silhouette width. Plos one. 2024 Jun 7;19(6):e0300358.

Minor Comment 4

The results (percentages) in Table 2 (Cluster demographics) probably need clarification: For example, the percentage of White Race in Table 1 is 77%, but the overall corresponding percentage in Table 2 is much higher (its lowest value is 83.3% for variable C8). Is this due to the "baseline visits" of Table 1 and the "all visits" for Table 2?

Response

Thank you for your observation. The difference in percentages between Table 1 and Table 2 is indeed due to the fact that Table 1 reflects demographic data based on baseline visits, while Table 2 represents demographic data across all visits. Since Table 2 accounts for multiple visits per individual, certain demographics, such as race, may appear more frequently due to the longitudinal nature of the dataset. To maintain consistency between the two tables, we have updated Table 2 to reflect baseline visits only, ensuring a more accurate comparison with the demographic percentages in Table 1. We have also clarified this in the caption of Table 2 and section 3.4 (“subtype demographics”) in the revised manuscript. 

Comments from Reviewer: 2

The authors argue that the growth of Electronic Health Records has opened new data-driven approaches to the heterogeneous trajectories associated with dementia. In their view, existing approaches have been limited by an emphasis on expensive neuroimaging and single time-point assessment. They, conversely, investigate the possibilities of trajectories in more widespread cognitive assessment tasks. Using the health records of visitors to a university memory clinic over a number of years, they first use unsupervised learning to identify different clusters of scores on a multidimensional assessment instrument and then recreate trajectories in terms of transitional probabilities between these clusters.

I think the authors make a good case for the value of this project. If useful information on dementia trajectories can be extracted from widely used tests, it would provide an efficient way to ground better assessments of risk and (potentially) personalization of treatments.

I also applaud the authors' clear and concise explanation of their approach and procedures. The paper was to the point, focused, and enjoyable to read.

In terms of minor points, there were a couple of places where things could be explained a little further.

Response

Thank you for your thorough review and positive comments on the manuscript. We have responded in a point-by-point manner to all your comments below.

Minor Comment 1

The description of measures on page 5 is not easy to understand; I was unclear if the history / neurological examination, the cognitive assessment battery and the CDR were three separate things or not. The list on page 6 makes it much clearer, so I'd suggest moving some of this information up to the earlier section.

Response

Thank you for your feedback on the clarity of the measures described. We agree that the distinction between the history/neurological examination, cognitive assessment battery, and Clinical Dementia Rating (CDR) could be clearer in the earlier section. To address this, we have revised the description on page 5 (section 2.1; “data sources and study participants”) to better differentiate these components. Further, we have added information from the list of cognitive assessment scores and CDR components on page 6 (section 2.2.1; “feature selection for clustering”) to improve clarity. This should help readers understand the structure and distinctiveness of the different terminologies from the outset. 

Minor Comment 2

Similarly, describing the dataset in the materials section without giving sample sizes (which came in the Results section) threw me a bit. I think it would be more comprehensible if the size of the data is mentioned earlier.

Response

Thank you for pointing out the need for better placement of the sample size information. We agree that introducing the sample sizes earlier in the Methods (section 2.1; “data sources and study participants”) would enhance the comprehensibility of the dataset description. We have added the sample size information in section 2.1 to provide a clearer context for readers as they proceed through the manuscript.

Minor Comment 3

The data period on page 5 seems to be five years, but on page 8 is described as six years.

Response

We appreciate your attention to this detail. The discrepancy in the data period was unintentional, and we will correct it to ensure consistency. The accurate data period is five years. We have updated both instances in the manuscript (section 2.1; “data sources and study participants” and section 2.4; “clustering on longitudinal visits”) to reflect the correct time period of the data.

Minor Comment 4

The rationales for clustering on CDR components (page 10) and for choosing 10 as the optimal number of clusters (page 11) could be filled out a bit more. Right now a few considerations are given, but I didn't quite understand what the deciding factor was in these decisions.

Response

Thank you for your feedback. Here is a brief explanation/justification of the rationales for selecting CDR components as the optimal features for clustering and choosing K=10 as the optimal number of clusters.

CDR components as the optimal feature set for clustering: 

The choice behind selecting CDR components as the optimal set of features for clustering can be justified in a both qualitative and quantitative way. 

Qualitative: We qualitatively analyzed the T-SNE distribution of two feature sets (CDR components and cognitive assessment scores) to determine the optimal input features for clustering. Cognitive scores created a gradient aligned with global CDR but did not produce distinct clusters (Fig 2A). In contrast, the CDR components produced well-separated clusters without significant overlap (Fig 2B), making them the most suitab

---

## [Decision Letter · Decision Letter 1]

24 Oct 2024

Examining heterogeneity in dementia using data-driven unsupervised clustering of cognitive profiles

PONE-D-24-28122R1

Dear Dr. KUMAR,

We’re pleased to inform you that your manuscript has been judged scientifically suitable for publication and will be formally accepted for publication once it meets all outstanding technical requirements.

Kind regards,

Matthew Cserhati, Ph.D

Academic Editor

PLOS ONE

Additional Editor Comments (optional):

Reviewers' comments:

Reviewer's Responses to Questions

**Comments to the Author**

1. If the authors have adequately addressed your comments raised in a previous round of review and you feel that this manuscript is now acceptable for publication, you may indicate that here to bypass the “Comments to the Author” section, enter your conflict of interest statement in the “Confidential to Editor” section, and submit your "Accept" recommendation.

Reviewer #3: All comments have been addressed

2. Is the manuscript technically sound, and do the data support the conclusions?

Reviewer #3: Yes

3. Has the statistical analysis been performed appropriately and rigorously? 

Reviewer #3: Yes

4. Have the authors made all data underlying the findings in their manuscript fully available?

Reviewer #3: Yes

5. Is the manuscript presented in an intelligible fashion and written in standard English?

Reviewer #3: Yes

6. Review Comments to the Author

Reviewer #3: (No Response)

7. PLOS authors have the option to publish the peer review history of their article (what does this mean?). If published, this will include your full peer review and any attached files.

Reviewer #3: No

---

## [Editor Report · Acceptance letter]

4 Nov 2024

PONE-D-24-28122R1 

PLOS ONE

Dear Dr. KUMAR, 

I'm pleased to inform you that your manuscript has been deemed suitable for publication in PLOS ONE. Congratulations! Your manuscript is now being handed over to our production team.

Kind regards, 

on behalf of

Dr. Matthew Cserhati 

Academic Editor

PLOS ONE